# The Spirocyclic Imine from a Marine Benthic Dinoflagellate, Portimine, Is a Potent Anti-Human Immunodeficiency Virus Type 1 Therapeutic Lead Compound

**DOI:** 10.3390/md17090495

**Published:** 2019-08-24

**Authors:** Mai Izumida, Koushirou Suga, Fumito Ishibashi, Yoshinao Kubo

**Affiliations:** 1Department of Clinical Medicine, Institute of Tropical Medicine, Nagasaki University, Nagasaki 852-8523, Japan; 2Department of Community Medicine, Graduate School of Biomedical Sciences, Nagasaki University, Nagasaki 852-8523, Japan; 3Graduate School of Fisheries and Environmental Sciences, Nagasaki University, Nagasaki 852-8521, Japan; 4Program for Nurturing Global Leaders in Tropical Medicine and Emerging Communicable Diseases, Graduate School of Biomedical Sciences, Nagasaki University, Nagasaki 852-8523, Japan

**Keywords:** portimine, *Vulcanodinium rugosum*, HIV-1, reverse transcriptase

## Abstract

In this study, we aimed to find chemicals from lower sea animals with defensive effects against human immunodeficiency virus type 1 (HIV-1). A library of marine natural products consisting of 80 compounds was screened for activity against HIV-1 infection using a luciferase-encoding HIV-1 vector. We identified five compounds that decreased luciferase activity in the vector-inoculated cells. In particular, portimine, isolated from the benthic dinoflagellate *Vulcanodinium rugosum*, exhibited significant anti-HIV-1 activity. Portimine inhibited viral infection with an 50% inhibitory concentration (IC_50_) value of 4.1 nM and had no cytotoxic effect on the host cells at concentrations less than 200 nM. Portimine also inhibited vesicular stomatitis virus glycoprotein (VSV-G)-pseudotyped HIV-1 vector infection. This result suggested that portimine mainly targeted HIV-1 Gag or Pol protein. To analyse which replication steps portimine affects, luciferase sequences were amplified by semi-quantitative PCR in total DNA. This analysis revealed that portimine inhibits HIV-1 vector infection before or at the reverse transcription step. Portimine has also been shown to have a direct effect on reverse transcriptase using an in vitro reverse transcriptase assay. Portimine efficiently inhibited HIV-1 replication and is a potent lead compound for developing novel therapeutic drugs against HIV-1-induced diseases.

## 1. Introduction

Various therapeutic drugs against human immunodeficiency virus type 1 (HIV-1) have been developed, although almost all of these drugs target viral enzymes. In general, initial HIV-1 regimens included three HIV-1 medicines from two or more drug classes. Usually, two types of nucleoside reverse transcriptase inhibitors are used in combination with a protease inhibitor, a nonnucleoside reverse transcriptase inhibitor or an integrase inhibitor. All of these drugs suppress HIV-1 enzymes. Because of the high mutation rate of viral replication, viral variants resistant to the drugs easily appear during drug treatment [1]. To resolve this problem, novel types of drugs are needed.

Hosts have evolved and developed many defence factors involved in innate immunity to protect themselves from viruses. Viruses have also evolved to obtain mechanisms to overcome these host defence factors. Human viruses can efficiently replicate in humans by overcoming human defence factors. For examples, SAM domain- and HD domain-containing protein 1 (SAMHD1) inhibits HIV-1 infection of dendritic cells [2,3] and T cells [4,5] by reducing the cellular deoxynucleoside triphosphate (dNTP) concentration to a level at which the viral reverse transcriptase cannot function [2,6]. HIV-2 developed Vpx to counteract the antiviral activity of SAMHD1.

Thus, it may be difficult to isolate host defence factors against human viruses from humans. Therefore, we tried to isolate defence factors against human viruses from species other than human. In particular, lower animals that do not have acquired immunity must protect themselves from viruses through only innate immunity, including defence factors. It has been reported that 10^6^–10^8^ virus particles exist per 1 mL of sea water and probably infect all marine life, from bacteria to whales [7]. These reports prompted us to speculate that lower sea animals have developed many antiviral factors. Thus, marine life forms are important sources of structurally diverse and biologically active secondary metabolites, several of which have inspired the development of new classes of therapeutic agents [8]. Indeed, many antiviral compounds have been already isolated from marine lower animals [9,10,11,12,13,14,15] (for review, [16,17,18]). In this study, we aimed to isolate novel defence chemicals against HIV-1 from lower sea animals and to elucidate the molecular mechanism for the inhibition of HIV-1 replication. As the first step, we screened a library of marine natural products consisting of 80 compounds isolated from marine organisms. We found that portimine, isolated from the benthic dinoflagellate *Vulcanodinium rugosum*, exhibited significant antiviral activity against HIV-1.

## 2. Results

### 2.1. Screening of 80 Compounds Extracted from Lower Sea Animals for the Inhibition of HIV-1 Vector Infection

We commercially obtained a library containing 80 compounds extracted from lower sea animals (Figure 1) and measured the impact of each compound on HIV-1 vector infection. These chemicals have been already dissolved in dimethyl sulfoxide (DMSO). First, we constructed a replication-defective HIV-1 vector expressing the luciferase gene [19]. HeLa cells expressing CD4 (HeLaCD4) were pretreated with each compound at 2 μM for 5 h and washed with phosphate buffered saline (PBS). The treated cells were inoculated with the vector, and the luciferase activities of the inoculated cells were measured (Figure 1). We identified five compounds, portimine, halichondramide, kabiramide B, swinholide A and 2-bromoaldisine, which decreased luciferase activity (Figure 1). Next, we analysed the effects of these five compounds at 200 nM concentration on HIV-1 vector infection. In 2-bromoaldisine- and portimine-treated cells, luciferase activity was decreased to 1/3 and 1/250 compared with control (DMSO-treated cells), respectively (Figure 1). Halichondramide, kabiramide B and swinholide A were excluded from the subsequent experiments, because they did not inhibit the vector infection at the 200 nM concentration. 

### 2.2. Bromoaldisine Inhibited HIV-1 Vector Infection

2-Bromoaldisine (Figure 2A) inhibited HIV-1 vector infection to 1/3 at 200 nM in a 96-well plate. Then, we analyzed its effect on cell viability. HeLaCD4 cells were treated with 2-bromoaldisine for 5 h, and stained with trypan blue. Numbers of unstained live cells were not changed by the treatment even at 200 nM (Figure 2B), showing that 2-bromoaldisine has no cytotoxicity. 

To measure 50% inhibitory concentration (IC_50_) of 2-bromoaldisine to inhibit HIV-1 Env-mediated infection, HeLaCD4 cells were pretreated with 2-bromoaldisine for 5 h at various concentrations, washed with PBS, and inoculated with luciferase-encoding, HIV-1 Env-carrying HIV-1 vector in a six-well plate. Luciferase activities of the inoculated cells were measured. Treatment at 50 nM did not affect the infection, but at 100 nM, HIV-1 Env-mediated infection was reduced to 1/10 (Figure 2C). The IC_50_ was about 76.0 nM. 2-Bromoaldisine also inhibited vesicular stomatitis virus glycoprotein (VSV-G)-pseudotyped HIV-1 vector infection (Figure 2D). The IC_50_ against VSV-G-mediated infection was similarly 87.6 nM.

### 2.3. Portimine Inhibited Retrovirus Vector Infection at Low Concentration

As portimine (Figure 3A) more efficiently inhibited HIV-1 Env-mediated infection than 2-bromoaldisine, we analyzed the inhibitory effect of portimine in detail. First, we assessed its cytotoxicity, but portimine showed no effect on cell viability (Figure 3B). This result indicated that the inhibitory effect on HIV-1 vector infection was not a result of cytotoxicity. 

To measure the IC_50_ of portimine to inhibit HIV-1 vector infection, HeLaCD4 cells were treated with portimine at different concentrations (Figure 4A). Portimine IC_50_ was approximated as 4.1 nM.

### 2.4. Portimine Inhibited Retrovirus Infection Independent of Viral Envelope Protein

To identify whether the inhibitory effect of portimine on HIV-1 vector infection is envelope-dependent, we constructed a VSV-G envelope pseudotyped HIV-1 vector encoding luciferase. Portimine 1 nM inhibited VSV-G-mediated infection to about 1/2 compared with DMSO treatment (Figure 4B). Portimine IC_50_ to inhibit VSV-G-pseudotyped HIV-1 vector infection was 1.1 nM. These results indicate that portimine inhibits both infections mediated by HIV-1 Env and VSV-G, suggesting that portimine mainly targets HIV-1 Gag or Pol protein. 

### 2.5. Portimine Inhibited Reverse Transcriptase

To analyse if portimine inhibits the reverse transcription step in the HIV-1 replication cycle, portimine-treated HeLaCD4 cells were inoculated with HIV-1 (JD34) Env or VSV-G-carrying HIV-1 vector encoding luciferase, and total DNA was extracted from the cells 5 h after the inoculation. Luciferase sequences were amplified by semi-quantitative PCR to measure the reverse transcription product. Reverse transcription generally takes place around 5 h after infection. The products were decreased by portimine (Figure 5A). This analysis revealed that portimine inhibits HIV-1 vector infection before or at the reverse transcription step.

Next, we investigated whether portimine directly inhibits reverse transcriptase (RT). Activity of purified HIV-1 RT was measured in the presence or absence of portimine using the reverse transcriptase assay. Portimine at 100 nM inhibited RT activity to about one-third compared with DMSO (Figure 5B). The inhibitory effect of portimine on RT was compared with that of efavirenz, a nonnucleoside analogue reverse transcriptase inhibitor (NNRTI). Efavirenz at 5 nM inhibited RT activity to one-tenth. Portimine at 50 nM inhibited RT activity to about one-fourth but, at 5–25 nM, inhibited RT activity to 60% (Figure 5B). This result suggested that portimine had a direct effect on RT and that it is less potent than efavirenz.

### 2.6. Portimine Inhibited HIV-1 Replication

In the previous experiments, replication-defective HIV-1 vector was used. We next evaluated whether HIV-1 replication was inhibited by portimine. MAGIC5A cells [20] were infected with the laboratory-adopted NL4-3 [21] or the primarily isolated 93JP-NH1 strain [22] of HIV-1, and we measured p24 amounts in culture supernatants using ELISA assay 0, 3 and 6 days after the inoculation. MAGIC5A cells were constructed from HeLa cells expressing the HIV-1 receptors CD4, C-C motif chemokine receptor 5 (CCR5), and C-X-C motif chemokine receptor 4 (CXCR4). Thus, the NL4-3 and 93JP-NH1 viruses could replicate in this cell line. The p24 amounts in culture supernatants 3 days after the NL4-3 virus inoculation in the portimine-treated cells (about 230 ng/mL) were similar to those in the untreated cells (150 ng/mL). The amounts of p24 in the NL4-3 virus-inoculated cells (1100 ng/mL) were decreased by portimine at 1 and 5 nM to one-fifth at 6 days after inoculation (250 ng/mL) (Figure 6A). Portimine treatment at 1 and 5 nM for 6 days did not affect the numbers of MAGIC5 cells, but portimine at 10 nM reduced cell number compared with DMSO treatment (data not shown). However, treatment with portimine at 12.5 nM for 5 h had no effect on cell viability (Figure 3). Although 60 ng/mL p24 was detected in culture supernatant from the 93JP-NH1 virus-inoculated cells 3 days after the inoculation, the p24 protein was not detected in the presence of portimine (Figure 6B). Portimine reduced p24 levels to about one-half at 6 days after inoculation (230 to 110 ng/mL). These results showed that portimine efficiently inhibited HIV-1 replication. 

## 3. Discussion

In this study, we found that portimine isolated from *Vulcanodinium rugosum* inhibited HIV-1 infection at nM order without cytotoxicity to HeLa cells partly by suppressing HIV-1 RT. Portimine is a potent lead compound for development of novel anti-HIV-1 drugs. 

Consistent with our results, portimine has exhibited low acute toxicity in mice [23]. The LD_50_ of portimine following intraperitoneal administration to mice was 1570 μg/kg. A previous report also indicated that portimine has much lower toxicity than the other cyclic imine toxin [24]. In contrast, several studies reported that portimine exhibits toxicity in several cells in culture [25,26]. Similarly, in our study, portimine was toxic to peripheral blood mononuclear cells even at 1 nM (data not shown). It has been reported that portimine induces apoptosis by rapid caspase activation [23]. These results indicate that portimine-mediated cytotoxicity is cell line-dependent. To understand why portimine has cytotoxicity on T cells but not on HeLa cells, further study is needed.

As portimine had cytotoxicity on T cells and PBMC, therapeutic index of portimine is thought to be low. However, the partial synthesis of portimine has been reported [27], and structural analogues of portimine may be constructed for biological evaluation. By analysing the anti-HIV-1 activity and the cytotoxicity of these portimine analogues, the relation of chemical structures with these two actions of portimine could be elucidated, and thus, we may find compounds that efficiently inhibit HIV-1 replication, but not cell viability. Finally, portimine is a potent lead compound for development of novel anti-HIV-1 drugs. 

Replication-defective HIV-1 vector infection was decreased to one-tenth by portimine at 12.5 nM. The amounts of HIV-1 p24 protein in culture supernatants from cells inoculated with replication-competent NL4-3 HIV-1 were attenuated to one-fifth by portimine at 1 nM. However, HIV-1 RT activity was suppressed to one-fourth by portimine at 50 nM. HIV-1 replication was more efficiently inhibited by portimine than RT activity was. These results suggest that portimine suppresses other step(s) of the HIV-1 replication cycle in addition to RT. Thus, the anti-RT activity of portimine was lower than that of efavirenz, but total antiviral activity of portimine is comparable.

In this study, we additionally found that 2-bromoaldisine isolated from marine sponge inhibits HIV-1 vector infection, although its anti-HIV-1 activity was lower than that of portimine. 2-Bromoaldisine is known to inhibit the Raf/MEK/MAPK pathway [28] that is required for HIV-1 infection [29] and expression [30]. Thus, 2-bromoaldisine might inhibit HIV-1 infection by suppressing Raf/MEK/MAPK signaling.

Halichondramide, swinholide A, and kabiramide B inhibited HIV-1 vector infection at 2 μM, but enhanced at 200 nM. It has been reported that halichondramide has anti-proliferative effect by suppression of mTOR [31]. The inhibition of HIV-1 vector infection by halichondramide at 2 μM might be mediated by its cell growth inhibition. In contrast, rapamycin, another mTOR inhibitor, enhances HIV-1 vector infection [32]. Thus, the treatment of halichondramide at 200 nM might promote HIV-1 vector infection by suppressing mTOR. It has been shown that swinholide A has cytotoxic effect by severing actin filaments [33]. Rearrangement of actin cytoskeleton is involved in HIV-1 entry to host cells [34]. Swinholide A treatment at 2 μM and 200 nM might inhibit HIV-1 vector infection by its cytotoxicity, and enhance by modulating actin filaments, respectively. Biological function of kabiramide B has not been elucidated yet.

It is thought that shellfishes obtain imine toxins from dinoflagellates to protect themselves from their natural enemies. Shellfishes containing such marine toxins sometimes cause food poisoning in humans. Marine toxins induce cell death at nM order, where portimine inhibits HIV-1 replication. Thus, *Vulcanodinium rugosum* may have portimine at the concentration that inhibits HIV-1 replication. The lower sea animal may obtain portimine to protect itself from viruses

In summary, we found that portimine inhibits HIV-1 replication partly by suppressing RT activity without cytotoxicity at nM order. *Vulcanodinium rugosum* may have portimine to protect itself from viruses. Portimine is a successful lead compound used for developing novel therapeutic drugs against HIV-1-induced diseases.

## 4. Materials and Methods

### 4.1. Cell Lines and Plasmids

HeLa cells were provided by Dr. H. Sato. HeLaCD4 cells were constructed [35] and maintained in our laboratory. MAGIC5 cells were obtained from Dr. H. Sato [25]. Cells used in this study were cultured in Dulbecco’s Modified Eagle Medium (FUJIFILM Wako, Osaka, Japan) with 8% foetal bovine serum (Sigma-Aldrich, St. Louis, MO) and 1% penicillin–streptomycin (Sigma-Aldrich, St. Louis, MO).

The Luciferase-encoding HIV-1 vector genome expression plasmid was provided by Dr. N. Landau through the AIDS Research and Reference Reagent Program, National Institute of Allergic and Infectious Diseases, National Institute of Health, United States [19]. This plasmid also encodes HIV-1 Gag-Pol, but not Env. The HIV-1 JD34 Env expression plasmid was kindly provided by Dr. U. Hazan [36]. The VSV-G expression plasmid was obtained from Dr. L. J. Chang through the AIDS Research and Reference Reagent Program, NIAID, NIH, United States [37]. The infectious molecular clones, HIV-1 NL4-3 and 93JP-NH1, were kindly provided by Dr. A. Adachi [24] and Dr. H. Sato [25], respectively.

### 4.2. HIV-1 Vector

HEK293T cells were transfected by the luciferase-encoding HIV-1 vector genome expression plasmid together with the indicated viral envelope protein expression plasmid, HIV-1 JD34 or VSV-G, using the FuGENE transfection reagent (Promega, Madison, WI, USA). Culture supernatants of the transfected cells were changed to fresh media 24 h after the transfection, and cells continued to be cultured for a further 24 h. Culture supernatants of the transfected cells were used to inoculate the target cells.

### 4.3. Chemical Library

HeLaCD4 cells were treated with each of the 80 compounds from lower sea animals (OP Bio Factory, Okinawa, Japan). Five hours after the treatment, cells were washed with PBS twice and were inoculated with luciferase-encoding HIV-1 vector. The cells were washed with PBS, and luciferase activity was measured using the luciferase assay system (Promega, Madison, WI, USA) 48 h after the inoculation.

### 4.4. Semi-Quantitative PCR

HeLaCD4 cells were inoculated with HIV-1 vector encoding luciferase. Total DNA was isolated from the inoculated cells 5 h after the inoculation. PCR was performed using the Veriti Thermal Cycler (Applied Biosystems, Singapore) as follows. Samples were denatured at 98 °C for 15 s and annealed with primers at 65 °C for 30 s, and DNA synthesis was extended at 72 °C for 3 min. The reaction was repeated 30 times. The KOD FX DNA polymerase was purchased from TOYOBO (Osaka, Japan). Nucleotide sequences of the primers to amplify luciferase sequence are CCCCCAGAAGCAATTTCGTG and TTTCGGCAGCCTACCGTAGTG.

### 4.5. Reverse Transcriptase Assay

HIV-1 reverse transcriptase (RT) and efavirenz were obtained from BioAcademia (Osaka, Japan) and Tokyo Chemical Industry (Tokyo, Japan), respectively. RT was diluted 1/10^6^-fold with PBS. Diluted RT was incubated with substrates at 37 °C in the presence of portimine for 30 min. A colorimetric enzyme immunoassay was performed using the Reverse Transcriptase Assay Kit (Roche, Mannheim, Germany).

### 4.6. HIV-1 Replication

HEK293T cells were transfected by the plasmid encoding infectious HIV-1 NL4-3 or 93JP-NH1. MAGIC5 cells were inoculated with 100 μL of culture supernatant from the transfected cells in a 6-cm dish. Amounts of HIV-1 p24 were measured 0, 3 and 6 days after the inoculation using the p24 ELISA Kit (ZeptoMetrix, Buffalo, NY, USA).

### 4.7. Statistical Analysis

Differences between two groups of data were determined using Student’s *t*-test. Statistical significance was set at *p* < 0.05 for all tests.

## Figures and Tables

**Figure 1 marinedrugs-17-00495-f001:**
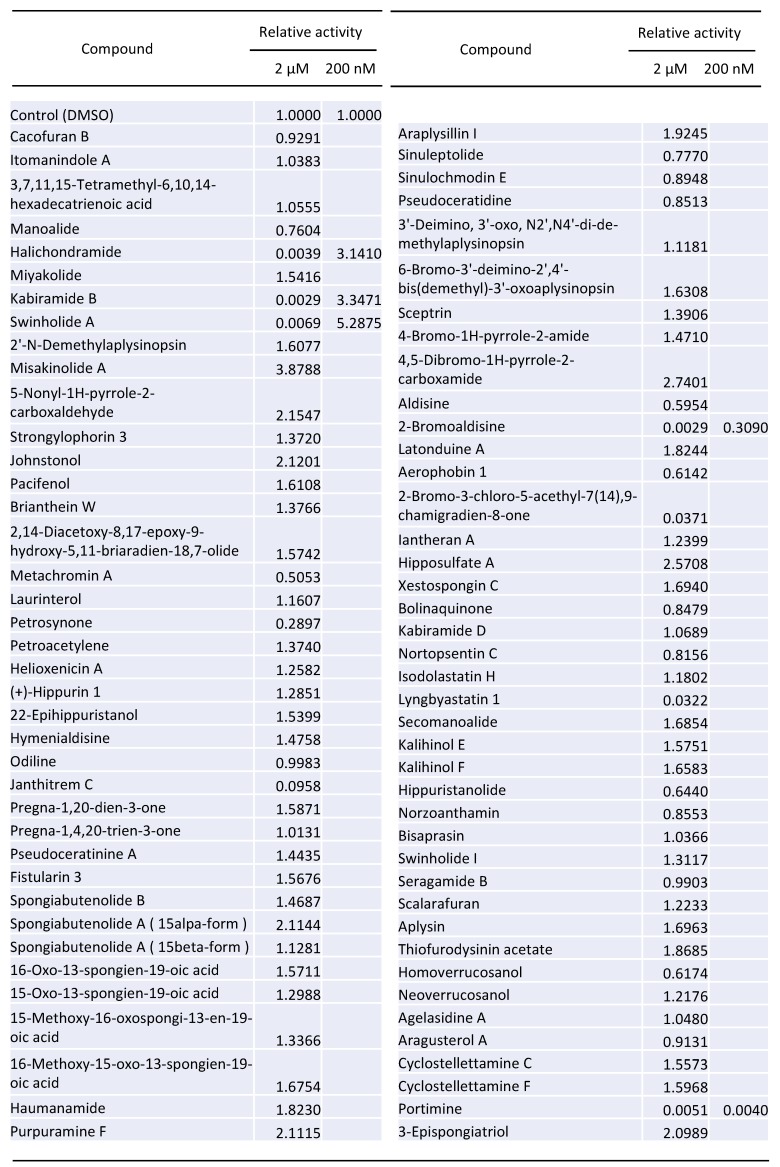
HIV-1 vector infection screening using luciferase assay of 80 marine natural products.

**Figure 2 marinedrugs-17-00495-f002:**
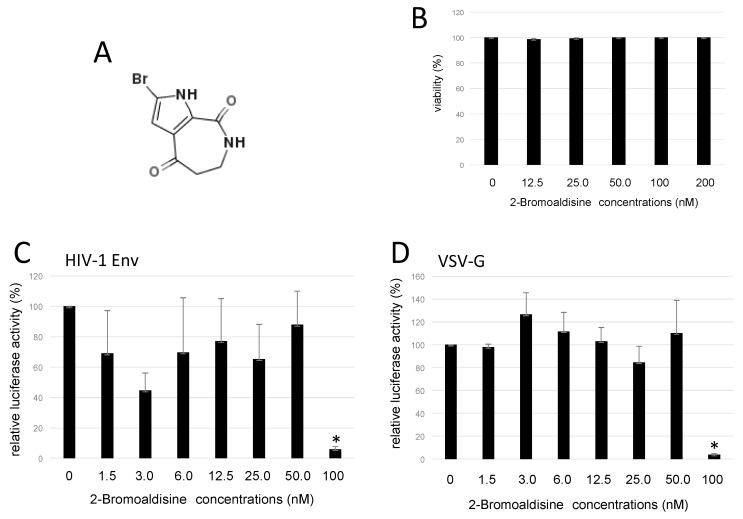
Effect of 2-bromoaldisine on cell viability and HIV-1 vector infection. (**A**) Chemical structure of 2-bromoaldisine. (**B**) HeLaCD4 cells were treated with 2-bromoaldisine for 5 h, and numbers of live cells were counted. (**C**) Treated cells were inoculated with HIV-1 Env-carrying HIV-1 vector. Luciferase activities of the inoculated cells were measured. (**D**) Treated cells were inoculated with VSV-G-pseudotyped HIV-1 vector. Error bars represent sample standard deviation from triplicate measurements. * *p* < 0.05 vs. control.

**Figure 3 marinedrugs-17-00495-f003:**
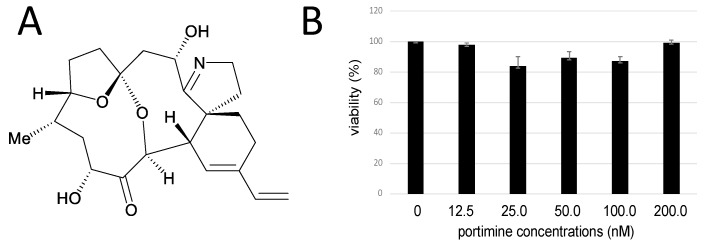
Cytotoxicity of portimine against HeLa cells expressing CD4 (HeLaCD4 cells). (**A**) Chemical structure of portimine. (**B**) HeLaCD4 cells were treated with various concentrations of portimine for 5 h. Ratios of alive cells to total cells were measured by trypan blue staining. Error bars represent sample standard deviation from triplicate measurements.

**Figure 4 marinedrugs-17-00495-f004:**
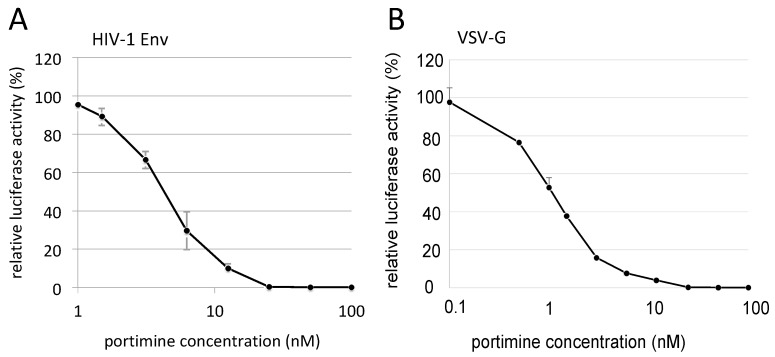
Dose-dependent inhibition of HIV-1 vector infection by portimine. (**A**) HeLaCD4 cells were treated with portimine at various concentrations and were inoculated with HIV-1 Env-carrying HIV-1 vector encoding luciferase. (**B**) Treated cells were inoculated with VSV-G-pseudotyped HIV-1 vector. Error bars represent sample standard deviation from triplicate measurements.

**Figure 5 marinedrugs-17-00495-f005:**
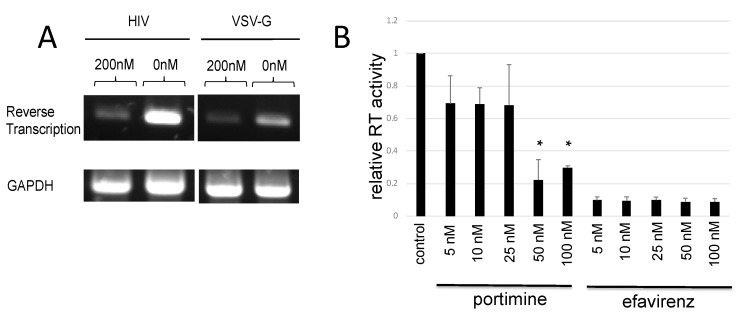
Portimine inhibits HIV-1 RT: (**A**) HeLaCD4 cells were treated with portimine at a concentration of 200 nM and washed twice with PBS 5 h after the treatment. HeLaCD4 cells were inoculated with HIV-1 vector encoding luciferase. DNA was extracted 5 h after the inoculation. (**B**) RT activity was measured in the presence of portimine or efavirenz. Error bars represent sample standard deviation from triplicate measurements. Significant results: * *p* < 0.05 vs. control.

**Figure 6 marinedrugs-17-00495-f006:**
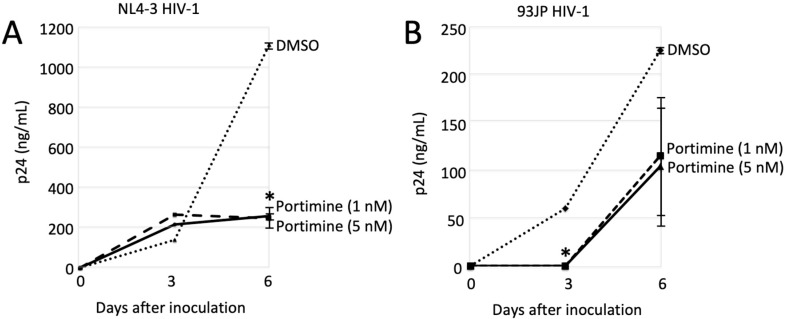
p24 amounts in culture supernatant from HIV-1-inoculated cells in the presence of portimine to MAGIC5A cells were inoculated with the NL4-3 (**A**) or 93JP-NH1 (**B**) strain of HIV-1. Error bars represent sample standard deviation from triplicate measurements. Significant results: * *p* < 0.05 vs. control (DMSO).

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
