# Peer review of "The Spirocyclic Imine from a Marine Benthic Dinoflagellate, Portimine, Is a Potent Anti-Human Immunodeficiency Virus Type 1 Therapeutic Lead Compound"

_marinedrugs, 2019, doi:10.3390/md17090495_

Round 1
Reviewer 1 Report
Izumida et al., in their report titled “The spirocyclic imine from a marine benthic dinoflagellate, portimine, is a potent anti-HIV-1 therapeutic lead compound” have screened 80 natural compounds extracted from lower sea animals and identified portimine as a potential compound that shows anti-HIV-1 activity. Portimine also only inhibited VSV-G pseudotyped HIV-1 suggesting the inhibition of portimine is independent of the viral envelope glycoprotein. They identified the inhibition by portimine is at the reverse transcription level as the RT products were decreased. Portimine has inhibited the replication of HIV-1 at concentrations of 5 nM. Based on their study authors conclude that lower sea animals may obtain portimine to protect itself from viruses.
Although this study provides some information on the natural compounds in the lower sea such as portimine have potential in protecting animals from viruses, there are several concerns need to be addressed.
Major concerns:
1. The antiviral activity of portimine is much lower compared to efavirenz.
2. Portimine reduced the cell numbers of MAGIC-5 cells at concentrations 10 nM when treated with longer time such as 6 days, and its IC50 concentration is 4.1 nM suggesting that the therapeutic index is very low for this compound.
3. What is IC50 of portimine against VSV-G pseudotyped HIV-1?
4. If replication of HIV-1 in the presence of portimine is carried out for a longer time, does resistance develop for this compound?
Minor points:
1. Reverse transcriptase is generally referred to as “RT” and NOT “RTase”.
2. Fig. 1B. How long HeLaCD4 cells were treated with portimine in this experiment?
3. Fig. 5A. What is p24 levels in the DMSO control at day 3?
4. At what concentration portimine shows toxicity to peripheral blood mononuclear cells?
Author Response
Answers to reviewer #1
Major concerns
Comment #1
As the reviewer mentioned, anti-RT activity of portimine was lower than that of efavirenz. However, HIV-1 replication was efficiently inhibited by portimine at 1 nM. Thus, total anti-HIV-1 activity of portimine is comparable to that of efavirenz (line 206-207).
Comment #2
As the reviewer mentioned, portimine had cytotoxicity to T cells and PBMC, and its therapeutic index is low. However, by synthesis of portimine analogues, we can construct novel chemicals with anti-HIV-1 activity but without cytotoxity. Portimine is a potent lead compound (line 194-200).
Comment #3
We evaluated IC50against VSV-G HIV-1 (line 126-127).
Comment #4
We measured p24 levels 12 days after the inoculation. The p24 amounts were very low both in the DMSO- and portimine-treated cells, as almost all cells died. It is known that HIV-1-infected cells die and p24 levels are reduced after all cells are infected, because some of HIV-1 proteins are toxic. On the other hand, portimine-treated cells were almost confluent 6 days after inoculation, but many syncytia were observed in DMSO-treated cells. The portimine-treated cells could be killed by over-confluent. Thus, it was thought that portimine-resistant viruses did not appear in this experiment.
Minor points
Comment #1
As the reviewer suggested, reverse transcriptase is referred to as RT, not RTase (line 144-151).
Comment #2
HeLaCD4 cells were treated with portimine for 5 h, same as in the inoculation experiment (Figure 2B legend, line 120).
Comment #3
Intact values of p24 (ng/mL) were added in the result section (line 165-176).
Comment #4
Growth of PBMC was significantly inhibited by portimine even at 1 nM (line 190).
Reviewer 2 Report
Dear authors
First of all , congratulations for the job.
An updated bibliography is missing due to are 25 % of bibliographic references older than 15 years
Best regrdas
Author Response
Answer to Reviewer #2
As the reviewer advised, many new references were added (ref. 10-19, 32-35).
Reviewer 3 Report
The manuscript by Mai Izumida et al., The spirocyclic imine from a marine benthic dinoflagellate, portimine, is a potent anti-HIV-1 therapeutic lead compound reports discovery of portimine as an anti-HIV compound from marine benthic dinoflagellate called Vulcanodinium rugosum. Overall, the data are interesting, are presented clearly and the manuscript is well written and except for some concerns that are discussed below. I would recommend this manuscript for publication in Marine Drugs, provided the authors reasonable address the following points.
#1. Page 3, lines 53-65: Thousands of novel compounds with antiviral activities have been isolated from natural sources, and some have been successfully developed for clinical use. Among them, arabinosyl nucleosides from a marine sponge Tethya cripta provided the basis for drug design of nucleoside analogs used in medicine today. After this discovery, much attention subsequently turned to marine organism-based drug discovery. In other words, more than half a century ago, it is known that there is a great relation between antivirals (including anti-HIVs) and marine products, so the present logical extension of the authors without this point includes the seemingly contradictory assertion.
#2. Pages 3-4, lines 85-89: The authors should show in set of anti-HIV activity and cytotoxicity, at least 5 chemicals discovered by first screening. Not each of these, but both, has useful meaning for readers.
#3. Table 1: The authors should mention about the abnormal values of > 1.0 at 200 nM in discussion section: for Halichondramide, Kabiramide B, and Swinholide A. Also, the authors should describe about the solubility for all 80 chemicals under assay condition. Do the chemicals dissolve completely in each cellular assay condition, especially at 2 µM?
#4. No information about semi-quantitative PCR are in Materials and Methods section. A minimum of experimental conditions, such as sequence of used primers, kind of Taq, and cycles and temperature of PCR, must be given in scientific papers.
[Minor]
- Fig. 4A: Reverse Transcription -> Reverse Transcriptas
Author Response
Answers to Reviewer #3
Comment #1
As the reviewer suggested, sentences about this issue was inserted (line 64-65), and some references were added.
Comment #2
Halichondramide, kabiramide B, and swinholide A at 200 nM did not reduce luciferase activity. This result suggests that these compounds do not have cytotoxity. Since 2-bromoaldisine inhibited vector infection at 200 nM, we analyzed its effect on cell viability and infection by HIV-1 Env- or VSV-G-carrying HIV-1 vector. The results (Figure 1) (line 87-99) and discussion (line 208-212) were added.
Comment #3
As the reviewer advised, discussion about halichondramide, swinholide A, and kabiramide B was added (line 213-222). We commercially obtained the chemical library. The chemicals have been already dissolved to 1 mg/ml in DMSO (line 74-75).
Comment #4
Protocol of semi-quantitative PCR was added to the Materials and Methods section (line 256-263).
Minor comment
As the reviewer suggested, the title of Figure 4 was changed (line 154).
Round 2
Reviewer 1 Report
Authors have addressed most of the comments. So the manuscript can be accepted for publication.